# *AdRAP2.3*, a Novel Ethylene Response Factor VII from *Actinidia deliciosa*, Enhances Waterlogging Resistance in Transgenic Tobacco through Improving Expression Levels of *PDC* and *ADH* Genes

**DOI:** 10.3390/ijms20051189

**Published:** 2019-03-08

**Authors:** De-Lin Pan, Gang Wang, Tao Wang, Zhan-Hui Jia, Zhong-Ren Guo, Ji-Yu Zhang

**Affiliations:** Institute of Botany, Jiangsu Province and Chinese Academy of Sciences, Nanjing 210014, China; PPxsperfect@163.com (D.-L.P.); wg20092011@163.com (G.W.); immmorer@163.com (T.W.); 13915954315@163.com (Z.-H.J.); zhongrenguo@cnbg.net (Z.-R.G.)

**Keywords:** ethylene-responsive factor, *Actinidia deliciosa*, *AdRAP2.3*, gene expression, waterlogging stress, regulation

## Abstract

APETALA2/ethylene-responsive factor superfamily (AP2/ERF) is a transcription factor involved in abiotic stresses, for instance, cold, drought, and low oxygen. In this study, a novel ethylene-responsive transcription factor named *AdRAP2.3* was isolated from *Actinidia*
*deliciosa* ‘Jinkui’. *AdRAP2.3* transcription levels in other reproductive organs except for the pistil were higher than those in the vegetative organs (root, stem, and leaf) in kiwi fruit. Plant hormones (Salicylic acid (SA), Methyl-jasmonate acid (MeJA), 1-Aminocyclopropanecarboxylic Acid (ACC), Abscisic acid (ABA)), abiotic stresses (waterlogging, heat, 4 °C and NaCl) and biotic stress (*Pseudomonas Syringae* pv. *Actinidiae*, *Psa*) could induce the expression of *AdRAP2.3* gene in kiwi fruit. Overexpression of the *AdRAP2.3* gene conferred waterlogging stress tolerance in transgenic tobacco plants. When completely submerged, the survival rate, fresh weight, and dry weight of transgenic tobacco lines were significantly higher than those of wile type (WT). Upon the roots being submerged, transgenic tobacco lines grew aerial roots earlier. Overexpression of *AdRAP2.3* in transgenic tobacco improved the pyruvate decarboxylase (PDC) and alcohol dehydrogenase (ADH) enzyme activities, and improved the expression levels of waterlogging mark genes *NtPDC*, *NtADH*, *NtHB1*, *NtHB2*, *NtPCO1*, and *NtPCO2* in roots under waterlogging treatment. Overall, these results demonstrated that *AdRAP2.3* might play an important role in resistance to waterlogging through regulation of *PDC* and *ADH* genes in kiwi fruit.

## 1. Introduction

The plant APETALA2/ethylene-responsive factor (AP2/ERF) superfamily is one of the largest plant transcription factor families participating in plant development and resistance to biotic and abiotic stresses [1,2,3,4]. Transcription factors are known as regulating plant stress responses through binding to *cis*-acting elements in the promoters of stress-related genes or interacting with other transcription factors [5]. AP2/ERF superfamily members contain 60‒70 highly conserved amino acids [6]. The AP2/ERF superfamily has been characterized based on either one or two AP2 domains: the AP2 subfamily has two AP2/ERF domains, the RAV subfamily has one AP2 domain and a B3 domain, C-repeat binding transcription factor/dehydrate responsive element binding factor (CBF/DREB) and the ERF subfamily have one AP2 domain [6]. ERF subfamily has conserved 14th alanine acid (A) and 19th aspartic acids (D) in the AP2/ERF domain. ERF subfamily transcription factors have been reported to be involved in biotic and abiotic stress [5]. The ERF subfamily was further divided into six small subgroups (V, VI, VII, VIII, IX, and X) based on the similarity of the amino acid sequences of their DNA-binding domain [1,7].

Members of the ERF subgroup VII have been identified to be involved in the hypoxic stress in *Arabidopsis thaliana*, including *RAP2.12*, *RAP2.2*, *RAP2.3*, *HRE1*, and *HRE2* [8,9]. The ERF VIIs were regulated by proteasome-mediated proteolysis via the oxygen-dependent branch of the N-end rule pathway [10,11]. Previous research showed that *RAP2.12* regulates central metabolic processes, for example, respiration, tricarboxylic acid (TCA) cycle, and amino acid metabolism [12]. *RAP2.2* has been proven to be induced by darkness, and overexpression of *RAP2.2* resulted in improving plant survival rate, increasing *ADH1* and *PDC1* expression levels, and increasing ADH and PDC enzyme activities [13]. In addition to *A. thaliana*, members of ERF subgroup VII have been found in other plants, for example, *Oryza sativa* [14] and *Petunia hybrida* [15].

The kiwi fruit is one of the most recently domesticated fruit crops. However, the majority of current kiwi fruit cultivars are susceptible to waterlogging stress. In many regions of China, especially in Eastern China, kiwi fruit plants suffer from excess rainfall during the summer rainy season. Waterlogging affects fruit yield severely and trees even die, which restricts the development of the kiwi fruit industry [16]. Kiwi fruit ‘Jinkui’ is tolerant to waterlogging, and understanding its waterlogging stress responses will be important for improving the tolerance of other kiwi fruit varieties. The transcriptome sequencing analysis in ‘Jinkui’ under waterlogging stress showed that the family of AP2/ERF transcription factors contained 14 upregulated and 14 downregulated members in the treatment samples. The transcription comp67160_0_seq1, which was a ERF VIIs member named *AdRAP2.3*, was strongly upregulated in roots of *Actinidia deliciosa* during the first 96 h after waterlogging treatment [17]. In order to analyze the function of the *AdRAP2.3* gene, the complete coding sequence (CDS) was cloned from the roots of ‘Jinkui’ under waterlogging stress according to the sequence comp67160_0_seq1 in this study. The expression patterns of *AdRAP2.3* in response to adverse stresses were investigated. The function of *AdRAP2.3* was further investigated by overexpression of *AdRAP2.3* in transgenic tobacco. 

## 2. Results

### 2.1. Cloning and Sequence Analysis of AdRAP2.3

The *AdRAP2.3* gene was isolated from *A. deliciosa* roots according to the sequence comp67160_0_seq1 [17]. The gene *AdRAP2.3* contains an 837 bp complete open reading frame (ORF), encoding 278 amino acids (Appendix A) with a predicted molecular weight (MW) of 31.27 kDa, a theoretical isoelectric point (pI) of 5.32, an instability index of 37.29, and an average hydrophilic coefficient of −0.831. The protein is stable and hydrophilic. The predicted *AdRAP2.3* has a typical conserved DNA-binding domain (AP2/ERF domain) of 58 amino acids. Moreover, AdRAP2.3 has conserved alanine (A) and aspartic acid (D) residues at the 14th and 19th positions in the AP2/ERF domain, as is characteristic of the ERF subfamily (Appendix A).

The analysis of the secondary structure of the amino acid sequence showed that the amino acid sequence encoded by *AdRAP2.3* contains the AP2 domain, which is position from 60 aa to 120 aa (Appendix A). The position 60 to 90 is DNA binding sites according to Appendix A. The three-dimensional structure of the protein encoded by the *AdRAP2.3* gene consists of one α-helix and three β-sheets according to Appendix A.

The 278 amino acid sequence predicted by the *AdRAP2.3* sequence was aligned with the amino acid sequences of other ERF VII subgroup proteins from *Arabidopsis*, *Oryza*, *Capsicum*, *Lycopersicon*, and *Actinidia*. The phylogenetic trees showed that RAP2.2 and RAP2.12 are closely related; SUB1C, SUB1B, and SUB1A-1 are closely related. PSR94738.1 is an ERF RAP2-3 like from *A. chinensis*, which is closely related to *AdRAP2.3* (Appendix A).

### 2.2. Expression Patterns of AdRAP2.3 in A. deliciosa

QRT-PCR assays were performed to investigate the expression pattern of *AdRAP2.3* in different organs and tissues of *A. deliciosa* (Figure 1). The expression level of *AdRAP2.3* is the highest in young fruit. *AdRAP2.3* transcription levels in other reproductive organs (except for pistils) were higher than those in the vegetative organs (root, stem, and leaf). 

The biotic stresses were considered to be correlated with the hormone signals [18,19]. Therefore, the expression patterns of *AdRAP2.3* under the treatment of SA, MeJA, ACC and ABA were analyzed by real-time PCR. As shown in Figure 2A, after SA treatment, *AdRAP2.3* mRNA accumulation increases with time and reaches a maximum at 12 h. Upon treatment with MeJA, the transcription level is induced obviously and reaches a peak at 4 h, then reduces at 48 h (Figure 2B). The transcription levels were induced, with an induction peak at 4 h, and then declines over time until 48 h in kiwi fruit after treatment with ACC (Figure 3C). *AdRAP2.3* mRNA accumulation increases with time and reaches a maximum at 12 h and then decreases in kiwi fruit after ABA treatment (Figure 2D). These results indicated that plant hormones (SA, MeJA, ACC, and ABA) could induce the expression of *AdRAP2.3* gene in kiwi fruit. 

Changes of *AdRAP2.3* transcription levels in *A. deliciosa* in response to various abiotic stresses were also analyzed by real-time PCR (Figure 3). As shown in Figure 3A, after 48 °C heat treatment, *AdRAP2.3* transcription level increases obviously at 2 h, reaching more than 30 times that in the control (0 h). The transcription level then decreases at 4 h (0 h). After 6 h recovery, the transcription level increases obviously. As shown in Figure 3B,C, after 4 °C and 0.2 M NaCl treatment, *AdRAP2.3* transcription level keeps going up during the first 48 h. *AdRAP2.3* mRNA accumulation increases about 400 times at 24 h and 600 times at 96 h after treatment with waterlogging (Figure 3D). Drought does not induce the expression of *AdRAP2.3* (Figure 3E). These results indicated that abiotic stresses including waterlogging, heat, low temperature (4 °C), and NaCl could induce the expression of *AdRAP2.3* gene in kiwi fruit. Compared with other environmental stresses, *AdRAP2.3* is strongly induced by waterlogging.

*Pseudomonas Syringae* pv. *actinidiae* (*Psa*) is known as a serious disease to kiwi fruit; it causes cankers, cracks, and a reddish bacterial ooze on trunks [20]. So we investigated the *AdRAP2.3* expression in response to *Psa* infection, and the expression levels were analyzed in phloem treated with *Psa* infection (Figure 4). The *AdRAP2.3* transcription level increased significantly at 24 and 96 h, indicating that the *AdRAP2.3* gene is involved in responses to *Psa* stress.

### 2.3. Overexpression of AdRAP2.3 Enhanced Waterlogging Tolerance in Transgenic Plants

To investigate the function of kiwi fruit *AdRAP2.3* gene, transgenic tobacco plants overexpression of *AdRAP2.3* were generated. A total of 40 independent transgenic lines (T_0_) were selected by hygromycin-resistance screening, and these transgenic lines were confirmed by β-glucuronidase (GUS) and PCR detection (Appendix A). The seeds from three representative *AdRAP2.3*-overexpressing lines (#1, #14 and #26) were selected for further functional analysis.

To investigated whether *AdRAP2.3* could increase waterlogging tolerance in transgenic tobacco plants. Seeds from three *AdRAP2.3*-overexpression lines and WT were planted on Murashige and Skoog (MS) medium for 10 days and then treated with completely submerged for 48 h (Figure 5). As shown in Figure 5A, the transgenic lines grow euphylla, the death rate of WT is significantly higher than those of transgenic lines (Figure 5B). Transgenic lines and WT were seeded on MS medium for 20 days and then treated with completely submerged for 7 d (Figure 6). The results showed that the transgenic lines grow better than WT after 5 d and 14 d recovery, and the fresh weight (Figure 6B) and dry weight (Figure 6C) of transgenic plants at recovery 14 d are significantly higher than those of WT respectively. To further verity that *AdRAP2.3* can confer the waterlogging tolerance of the overexpression lines, two-month-old transgenic plants and WT planted in soil were treated with root submerged (Figure 7). As shown in Figure 7B, after 7 d of treatment, transgenic lines grow aerial roots, but WT do not. These results showed that overexpression of *AdRAP2.3* could enhance the waterlogging tolerance in transgenic tobacco.

### 2.4. Physiological Changes in Transgenic Plants under Waterlogging Stress

Alcoholic fermentation through the coupled activity of PDC and ADH enzyme is of great importance in a plant’s ability to tolerate waterlogging [21,22,23]. All three transgenic lines showed improved resistance to waterlogging stress, and had similar results. The result of transgenic line #14 to improve waterlogging was a medium for all three transgenic lines. Transgenic line #14 was selected to explore the physiological mechanism of the waterlogging stress tolerance conferred by *AdRAP2.3* gene overexpression. PDC and ADH enzyme activities in root of WT and transgenic plants were measured in the control and in roots submerged for 24 d treatment (Figure 8). In the control, the PDC and ADH enzyme activities in the roots of transgenic lines are significantly higher than those of the WT. In terms of waterlogging stress, PDC and ADH enzyme activities in roots of transgenic lines are also significantly higher than those of WT. It can be concluded that overexpression of the *AdRAP2.3* gene can improve PDC and ADH enzyme activities.

### 2.5. Waterlogging-Related Genes Changes in Transgenic Plants under Waterlogging Stress

*ADH*, *PDC*, *HB1*, *HB2*, *PCO1*, and *PCO2* have been proved as marker genes in response to low oxygen stress [10,12]. There are no significantly difference between WT and transgenic lines on transcription levels of *NtPDC*, *NtADH*, *NtHB1*, *NtHB2*, and *NtPCO1* in roots under control condition (Figure 9). However, under waterlogging stress, the expression levels of *NtPDC*, *NtADH*, *NtHB1*, *NtHB2*, and *NtPCO1* in root of transgenic lines are significantly higher than those of WT (Figure 9). *NtPCO2* expression levels in roots are higher than those of WT under normal condition and waterlogging stress. Taken together, these results indicated overexpression *AdRAP2.3* can upregulate the expression levels of *NtPDC*, *NtADH*, *NtHB1*, *NtHB2, NtPCO1*, and *NtPCO2* under waterlogging stress. 

## 3. Discussion

### 3.1. Kiwi Fruit AdRAP2.3 Plays a Key Role in Resistance to Waterlogging Stress

Waterlogging or submergence caused O_2_ deprivation in the soil [24,25]. ERF members are important regulators of low oxygen tolerance extensively studied in many plants [8,10,11]. The ERF family plays a crucial role in the determination of survival of *Arabidopsis* and rice, which could reduce oxygen availability [11,26]. Previous studies showed that there are five ERF VIIs genes in *Arabidopsis*; two ERF VIIs genes (*HYPOXIA RESPONSIVE ERF1*/2) were greatly enhanced at the transcriptional and translational levels by O_2_ deprivation at multiple developmental stages, and the other three ERF VIIs genes were constitutively expressed (*RAP2.12*, *RAP2.2*, and *RAP2.3*) and further upregulated by darkness or ethylene in *A. thaliana* [11,27,28]. Transgenic *Arabidopsis* plants overexpressing *HRE1* showed an improved tolerance of anoxia [11]. The transcriptome sequencing analysis showed that there are 28 AP2/ERF transcription factors regulated by waterlogging in kiwi fruit. In this study, *A. deliciosa AdRAP2.3* gene was induced significantly by waterlogging, and overexpression of the *AdRAP2.3* gene conferred waterlogging stress tolerance in transgenic tobacco plants. When completely submerged, transgenic tobacco lines had a significantly higher survival rate, fresh weight, and dry weight compared to the WT. These results suggested that kiwi fruit *AdRAP2.3* plays a key role in resistance to waterlogging.

### 3.2. AdRAP2.3 Could Enhance Resistance to Waterlogging through Promoting Pneumatophore Production

Waterlogging may trigger different molecular and physiological disorders in plants. These include significant deterioration of plant water statues [29], decreases in leaf gas exchange variables, photoinhibition of photosystems (PSI and PSII) [30], and decreases in root system biomass. Using a light microscope, we see that the root anatomical structures of plants exhibit changes in root diameter, stele diameter, epidermal thickness, and xylem thickness under waterlogging stress [31]. In addition, the formation of aerenchym tissue is found after treated with waterlogging stress, including an increasing number of aerenchym cells and increasing length of aerenchym cells [31,32]. Aerenchyma formation under waterlogging stress is one of the most effective mechanisms to provide an adequate oxygen supply and overcome the stress-induced hypoxia imposed on plants [33]. The numerous pneumatophores contribute to morphological adaptations under waterlogging stress. Regardless of pneumatophores, stressed oil palm seedlings were able to adjust their leaf water status and gas exchange to cope with waterlogging [34]. Waterlogging subjects plant roots to an anoxic environment, limiting mitochondrial aerobic respiration and causing energy loss. In response to hypoxia stress, plants can temporarily compensate with anaerobic respiration [35]. In this study, the overexpression of *AdRAP2.3* induces pneumatophores under waterlogging in transgenic tobacco, suggesting that kiwi fruit *AdRAP2.3* could enhance resistance to waterlogging through promoting pneumatophore production.

### 3.3. Kiwi Fruit AdRAP2.3 Enhances Waterlogging Resistance in Transgenic Tobacco through Improving Expression Levels of PDC and ADH Genes

Three enzymes involved in anaerobic metabolic pathways, PDC, ADH, and lactate dehydrogenase (LDH), produce acetaldehyde, ethanol, and lactic acid, respectively. Although acetaldehyde and ethanol are thought to be harmful to plant cells, lactic acid is a major cause of root death through its induction of cytoplasmic acidification and pH reduction [36,37,38]. In this study, transgenic lines can improve PDC and ADH enzyme activities in the control. In response to long-term waterlogging stress, the PDC and ADH enzyme activities of WT decrease, whereas those of transgenic lines increase to sustain substrate-level adenosine triphosphate (ATP) production and promote hypoxia acclimation. These results indicate that *AdRAP2.3* promotes PDC and ADH enzyme activities.

Previously, a large number of high-throughput sequence (Tag-seq) analyses based on the Solexa Genome Analyzer platform were performed to analyze the gene expression profiling of plants under waterlogging stress. Differentially expressed genes (DEGs) are obtained, mainly linked to carbon metabolism, photosynthesis, reactive oxygen species generation/scavenging, and hormone synthesis/signaling [17,39,40,41,42]. Some waterlogging-responsive genes were isolated and identified, such as *RAP2.12*, *RAP2.2* [13], *HRE1*, *HRE2*, *AdPDC1* [43], *AdPDC2* [44], *AdADH1* [45], and *AdADH2* [45]. *RAP2.12* mRNA was described as sufficient for activating the anaerobic response in *Arabidopsis* [11]. The closest RAP2.12 homologue, RAP2.2, has been suggested to be functionally redundant in the induction of the anaerobic gene expression [13,46]. *HRE1* and *HRE2* are expressed at low levels under aerobic conditions and strongly upregulated by hypoxia [11]. Overexpression of *AdPDC1*, *AdPDC2*, *AdADH1*, and *AdADH2* in *Arabidopsis* enhanced waterlogging tolerance [43,44,45]. RAP2.12 was re-localized from the plasma membrane to the nucleus as O_2_ concentrations declined, with increased accumulation of hypoxia-responsive mRNAs, including PDC1 and hypoxia-responsive ERF1/2 (HRE1/2) [10]. In this study, we further analyzed the expression of the hypoxia marker genes *NtADH*, *NtPDC*, *NtHB1*, *NtHB2*, *NtPCO1*, and *NtPCO2* in the roots of WT and a transgenic line under submerged root stress, and the results showed that *AdRAP2.3* regulates these hypoxia marker genes mRNA levels under waterlogging conditions. Thus, kiwi fruit *AdRAP2.3* enhances waterlogging resistance in transgenic tobacco by improving expression levels of *PDC* and *ADH* genes.

In summary, the results showed that the increase of *AdRAP2.3* expression during waterlogging stress was much higher than that during other environmental stresses and that the kiwi fruit *AdRAP2.3* gene is required during waterlogging stress. Upon roots being submerged, transgenic tobacco lines grew aerial roots earlier than the WT. Overexpression of the *AdRAP2.3* gene in transgenic tobacco improved the activities of the PDC and ADH enzymes, and the expression levels of waterlogging mark genes *NtPDC* and *NtADH* in roots under waterlogging treatment. Overall, these results suggested that *AdRAP2.3* might play an important role in resistance to waterlogging through regulation of *PDC* and *ADH* genes in kiwi fruit. In future studies we will concentrate on introducing the *AdRAP2.3* gene into kiwi fruit varieties with poor waterlogging resistance, such as ‘Hongyang’, to determine whether this can enhance resistance to waterlogging.

## 4. Materials and Methods 

### 4.1. Plant Materials and Growth Conditions

The kiwi fruit cultivar ‘Jinkui’ was obtained from the Institute of Botany (Nanjing), Jiangsu Province and Chinese Academy of Sciences, China. The cutting seedlings from ‘Jinkui’ were grown in pots containing a 7:2:1 mixture of peat mold, vermiculite, and perlite in the greenhouse. Tissue culture seedlings of ‘Jinkui’ grew in a rooting medium (1/2 MS medium containing 0.6 mg/mL 1-naphthlcetic acid (NAA)) for one month. Shoots of ‘Jinkui’ were selected, surface sterilized, and grown in MS medium for one week. The conditions were: average temperature of 25 °C, relative air humidity of 60%, photoperiod of 16 h/8 h (light/dark), and quantum irradiance of 160 μmol m^−2^ s^−1^.

### 4.2. Treatments

To analyze the tissue-specific gene expression, different organs and tissues of ‘Jinkui’, including the root, stem, leaf, anthocaulus, petal, pistil, calyx, ovary, stamen, and fruitlet (20 days after full blossom, DAFB), were collected. To analyze gene expression patterns in response to hormones, 0.1 mM SA, 0.05 mM MeJA, 0.01 mM ACC, and 0.01 mM ABA were sprayed on the surface of the culture tissue seedlings leaves, and the leaves were collected at 0, 4, 12, and 48 h to analyze the gene expression pattern in response to stresses including heat, cold, salt, waterlogging, and drought. For heat and cold stress, the tissue seedlings were cultured at 48 °C for 0, 2, and 4 h, had a 23 °C recovery for 6 h, and then were exposed to 4 °C for 0, 4, 12, and 48 h, after which the leaves were collected. For salt stress, the seedlings were cultured in 28 cm × 14 cm × 14 cm containers with 0.2 M NaCl, and the leaves were collected at 0, 4, 12, and 48 h. For waterlogging, the seedlings were waterlogged in a 28 cm × 14 cm × 14 cm container filled with tap water to 2.5 cm above the level of the soil surface, and roots were sampled at 0, 24, 48, and 96 h. For drought, the cutting seedlings were cultured without watering for 14 d (the control was irrigated normally) and the leaves were sampled [45]. To analyze the gene expression pattern in response to *Psa*, bacterial cells were suspended in distilled water and adjusted to an OD_600_ = 0.2, then injected into the ‘Jinkui’ shoots, which were carved with a knife for 0, 24, 48, or 96 h, and then phloem from the shoots was collected [47]. Different samples were snap frozen in liquid nitrogen and stored at −80 °C for later experiments.

### 4.3. RNA Extraction and cDNA Synthesis

Total RNA was extracted from samples according to a method reported previously [48]. The purity and content of total RNA were detected by a spectrophotometer (Bruker BioSpin GmbH, Rheinstetten, Germany) and 1.0% agarose electrophoresis. The cDNA was achieved with a PrimeScript^TM^ RT reagent kit with gDNA Eraser (Perfect Real Time, TaKaRa, Cat. #RR047Q, Dalian, China), which could eliminate the residual DNA. The cDNA samples were diluted 1:10 with sterile double distilled water and stored at −20 °C before being used.

### 4.4. Gene Clone and Sequence Analysis

The cDNA sample from the ‘Jinkui’ root treatment with waterlogging for four days was used to amplify the complete CDS of *AdRAP2.3.* Gene-specific primers F1 and F2 were designed according to the comp67160_0_seq1 sequence (Appendix A) [17]. The open reading frame (ORF) was predicted by DNAstar 7.1.0(http://korwin-mikke.pl). Multiple alignments of the deduced amino acid sequence were performed using the the BioEdit software (v 7. 0. 5, Ibis Therapeutics, Carlsbad, CA, USA), and a phylogenetic tree was constructed by 1000 Bootstrap statistical tests with the Neighbor Joining (NJ) model by Mega 7.0 (https://www.megasoftware.net/). Molecular weight and isoelectric point were analyzed by Exspay (https://web.expasy.org/protparam/). The three-dimensional structure of *AdRAP2.3* was predicted by Swiss-plot (http://swissmodel.expasy.org/). 

### 4.5. Gene Expression Analysis Using Quantitative Real-Time PCR

The qRT-PCR was carried out on an Applied Biosystems 7300 Real Time PCR System (Applied Biosystems, Waltham, MA, USA) using TaKaRa Company SYBR Premix Ex TaqTM Ⅱ(Perfect Real Time, TaKaRa, code: DRR041A, Dalian, China). *AdActin* was used as internal reference gene to monitor cDNA abundance [49]. The quantitative PCR reaction system with specific primer contains 1 μL cDNA template, 10 μL 2 × SYBR Premix Ex Taq^TM^ Ⅱ, 0.3μL (10 pm) of each primer (Appendix A), and 8.4 μL ddH_2_O. The primer sequences used are listed in Appendix A. The reaction procedure is as follows: denaturation at 95 °C for 1 min, 95 °C for 20 s, 57 °C for 20 s, and 72 °C for 20 s; 45 cycles in total. Each sample set is repeated three times. After the reaction, the 2^−△△*C*t^ method was used to analyze the gene expression level. The statistical significance was assessed using SPSS 17.0 (SPSS Corp., Chicago, IL, USA).

### 4.6. Construction Binary Vector and Transformation of Tobacco

The coding sequence of *AdRAP2.3* was amplified by PCR using a specific primer pair (F1 and F2) modified to include 5′BamH I and 3′Sac I restriction sites. The fragment was inserted into the binary vector pCAMBIA 1301 under the control of the *Cauliflower mosaic virus* (CaMV35S) promoter. Then, the plasmid was introduced into *Agrobacterium tumefaciens*. Agrobacterium-mediated transformation of tobacco was performed by the leaf disc method [50]. The transgenic plants were detected by GUS staining and PCR analysis. The seeds of transgenic lines were harvested at the same stage and stored for subsequent analysis.

### 4.7. Phenotype Analysis of Transgenic Tobacco under Waterlogging Resistance 

Seeds from T_1_ progeny transgenic lines (#1, #14, and #26) and WT were surface sterilized and sown on the MS medium for 10 days; the death rate was measured after they were completely submerged for 48 h. Seedlings of transgenic lines and WT grew on MS medium for 20 d, the waterlogging treatment was performed for 7 d, and then seedlings were returned to normal growth conditions for 14 d. After 14 d recovery, the fresh weight and dry weight were measured. *Nicotiana tabacum* seedlings (#1, #14, #26, and WT) were transplanted into pots from the medium and grown in a greenhouse for two months under a 16/8-h light/dark cycle at 22/25 °C and 60% relative humidity. The pots were flooded in 28 cm × 14 cm × 14 cm containers filled with tap water to 2.5 cm above the level of the soil surface. The seedlings’ phenotypic changes during waterlogging stress assays were observed and photographed. The roots of transgenic lines and WT were collected after 24 d waterlogging and were later used for the measurement of enzyme activities and expression levels of downstream genes.

### 4.8. Measurement of Anaerobic Respiration and ADH and PDC Activities 

The enzyme ADH and PDC activities in roots of transgenic lines #14 and WT after 24 d waterlogging were measured spectrophotometrically by monitoring the oxidation of Nicotinamide adenine dinucleotide (NADH) at 340 nm [21]. The PDC assay reaction was carried out for 60 s at 25 °C. The ADH assay reaction was carried out for 10 min at 37 °C. One unit of PDC or ADH was defined as the amount of enzyme required to decompose 1 μmol of NADH per minute per gram fresh weight; at least 10 independent plants were evaluated in each test, and all tests were repeated three times.

### 4.9. Expression Analysis of Downstream Genes 

The expression levels of downstream genes in roots of transgenic line #14 and the WT after 24 d waterlogging were analyzed by qRT-PCR. The downstream genes *NtPDC*, *NtADH*, *NtHB1*, *NtHB2*, *NtPCO1*, and *NtPCO2* were acquired from NCBI (https://www.ncbi.nlm.nih.gov/). *NtTub* was used as an internal reference gene to monitor cDNA abundance [50]. The primers are listed in Appendix A. Statistically significant differences were calculated with SPSS 17.0 (SPSS Corp., Chicago, IL, USA). 

## Figures and Tables

**Figure 1 ijms-20-01189-f001:**
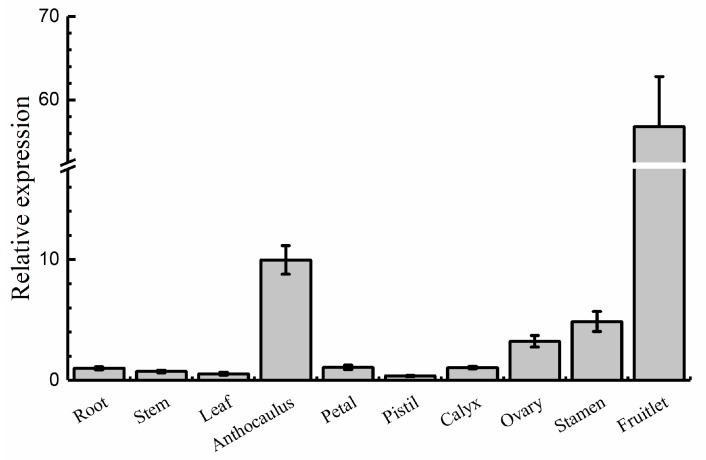
Expression analysis of gene *AdRAP2.3* in different kiwi fruit organs. *AdActin* transcription levels were used to normalize the samples. The mean and standard deviation were obtained from three independent experiments.

**Figure 2 ijms-20-01189-f002:**
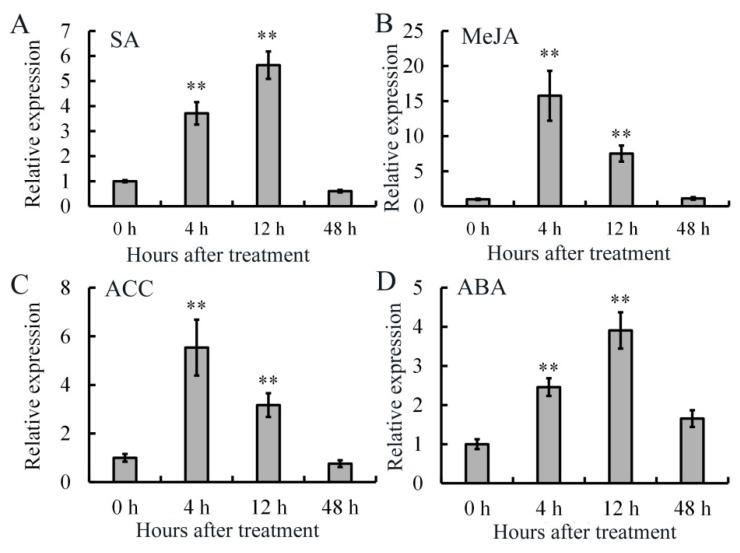
Expression analysis of *AdRAP2.3* in *A. deliciosa* ‘Jinkui’ leaves under different hormone treatments. (**A**) Salicylic acid (SA); (**B**) Methyl-jasmonate acid (MeJA); (**C**) 1-Aminocyclopropanecarboxylic Acid (ACC); (**D**) Abscisic acid (ABA). *AdActin* transcription levels were used to normalize the samples. The mean value and standard deviation were obtained from three independent experiments. The data represent averages ±Standard error (SE) of three biological repeats with three measurements per sample. ** indicates significant differences in comparison with the control at *p* < 0.01.

**Figure 3 ijms-20-01189-f003:**
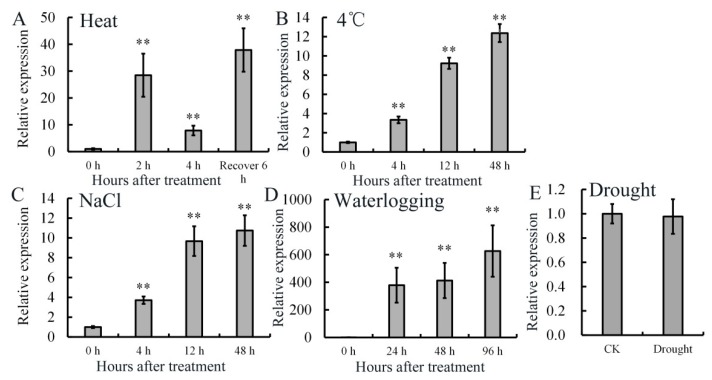
Expression analysis of *AdRAP2.3* gene under different abiotic stresses in kiwifruit. (**A**) Heat; (**B**) 4 °C; (**C**) NaCl; (**D**) waterlogging; (**E**) drought. *AdActin* transcription levels were used to normalize the samples. The mean value and standard deviation were obtained from three independent experiments. The data represent averages ± SE of three biological repeats with three measurements per sample. ** indicates significant differences in comparison with the control at *p* < 0.01.

**Figure 4 ijms-20-01189-f004:**
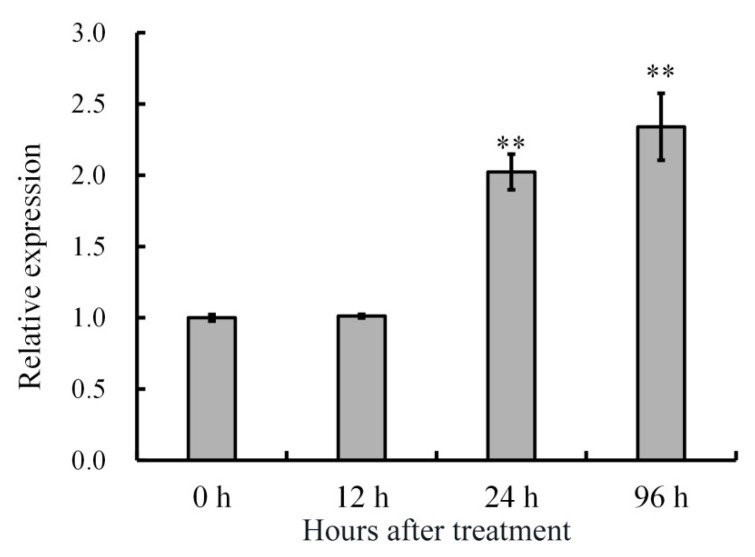
Expression analysis of *AdRAP2.3* gene under *Pseudomonas syringae* pv. *actinidiae* infection in kiwi fruit. *AdActin* transcription levels were used to normalize the samples. The mean value and standard deviation were obtained from three independent experiments. The data represent averages ± SE of three biological repeats with three measurements per sample. ** indicates significant differences in comparison with the control at *p* < 0.01.

**Figure 5 ijms-20-01189-f005:**
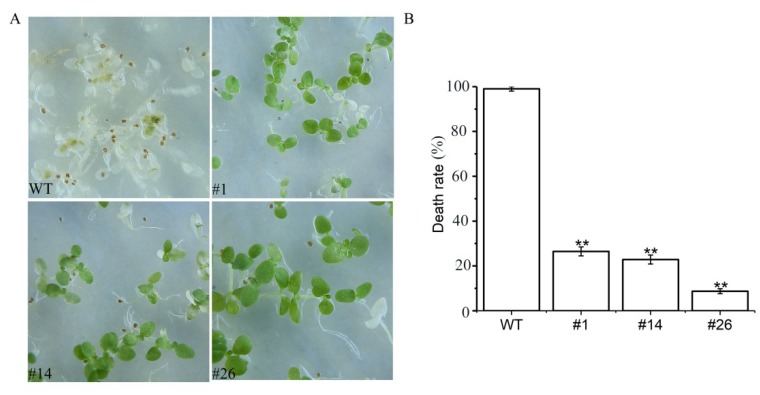
Phenotype (**A**) and death rate (**B**) in 10-day-old WT and transgenic tobacco line seedlings 48 h after they were completely submerged. WT: Wild type; #1,#14,#26: Transgenic tobacco lines. The data represent averages ± SE of three biological repeats with three measurements per sample. ** indicates significant differences in comparison with the WT at *p* < 0.01.

**Figure 6 ijms-20-01189-f006:**
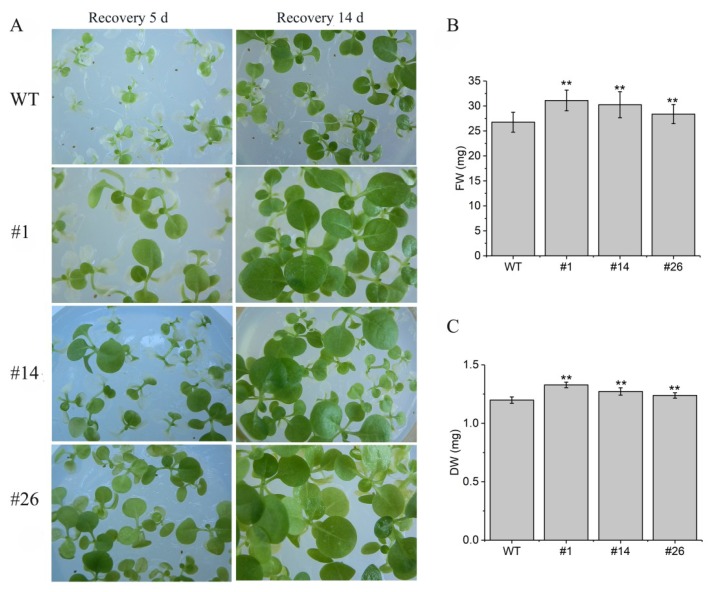
Phenotype, fresh weight (FW), and dry weight (DW) 7 d after completely submerged treatment for 20-day-old WT and transgenic tobacco lines seedlings. (**A**) Phenotype analysis; (**B**) fresh weight; (**C**) dry weight. WT: Wild type; #1, #14, #26: Transgenic tobacco lines. The data represent averages ± SE of three biological repeats with three measurements per sample. ** indicates significant differences in comparison with the WT at *p* < 0.01.

**Figure 7 ijms-20-01189-f007:**
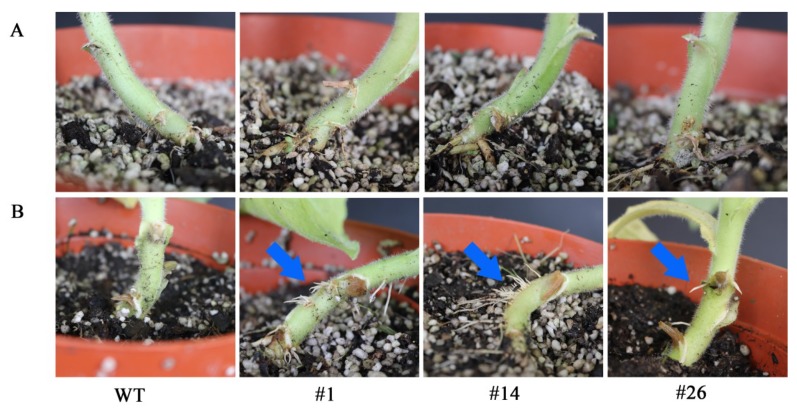
Morphological adaptations of two-month-old transgenic lines and WT plants were observed subjected to root submerged. (**A**) No waterlogging damage; (**B**) waterlogging for seven days. WT: Wild type; #1, #14, #26: Transgenic tobacco lines. Blue arrows indicated the aerial roots

**Figure 8 ijms-20-01189-f008:**
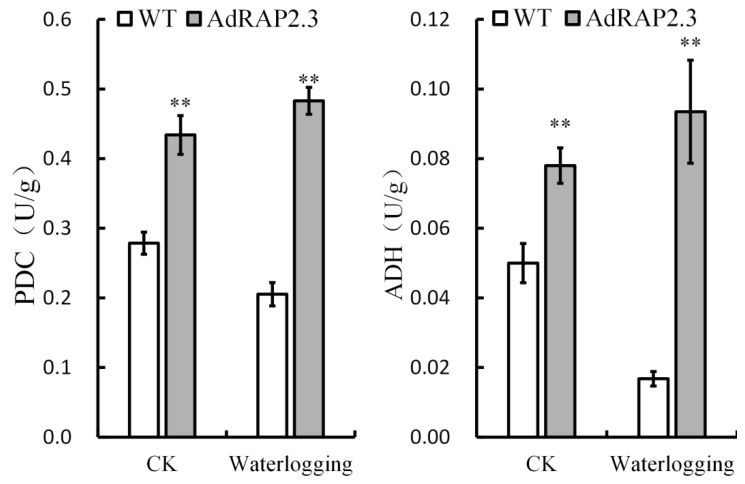
PDC and ADH enzyme activity measurements in roots of wild-type and AdRAP2.3 overexpressing lines #14. CK: normal growth conditions; WT: Wild type. The data represent averages ± SE of three biological repeats with three measurements per sample. ** indicates significant differences in comparison with the WT at *p* < 0.01.

**Figure 9 ijms-20-01189-f009:**
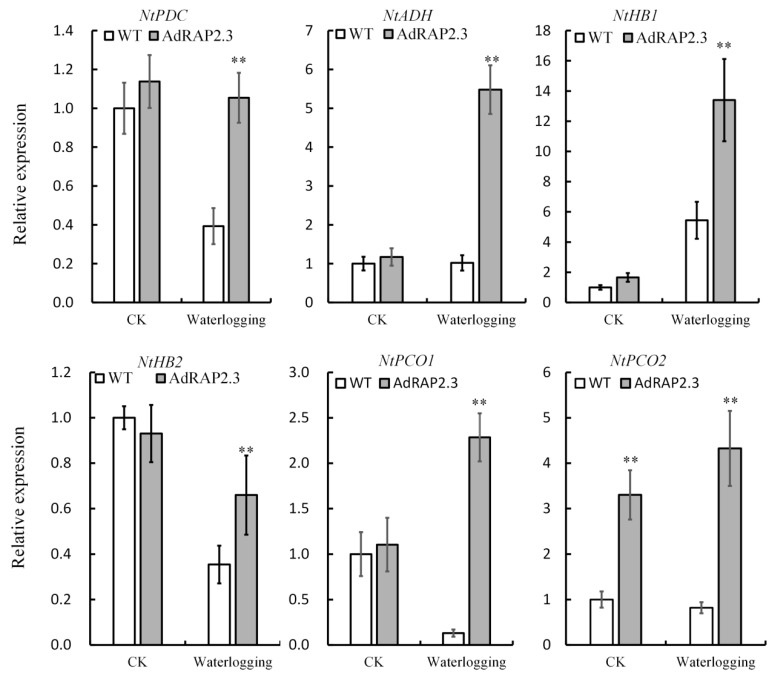
QRT-PCR analysis of the expression of the genes *NtPDC, NtADH, NtHB1, NtHB2, NtPCO1*, and *NtPCO2* in the root line of transgenic tobacco and the wild type during 24 d waterlogging stress. CK: Normal growth conditions; WT: Wild type. The data represent averages ± SE of three biological repeats with three measurements per sample. ** indicates significant differences in comparison with the WT at *p* < 0.01.

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
