# Peer review of "AdRAP2.3, a Novel Ethylene Response Factor VII from Actinidia deliciosa, Enhances Waterlogging Resistance in Transgenic Tobacco through Improving Expression Levels of PDC and ADH Genes"

_ijms, 2019, doi:10.3390/ijms20051189_

Reviewer 1 Report

The research was performed to elucidate the role of a member of AP2-ERF gene family in various stress conditions. It appears that AdRAP2.3 transcript showed high expression under heat and waterlogging conditions. Overall the research provides important information about involvement of this gene under waterlogging conditions, there are certain issues that can be addressed to improve the manuscript. 

1. How much productivity loss does water logging conditions cause in kiwifruit? I think introduction needs to be expanded in terms of the water logging stress and its role in limiting kiwifruit growth and production?

2. The gene was isolated from the roots of kiwifruit genotype "jinkui". Was this genotype water-logging tolerant? This information should be included in the introduction part. If this genotype is tolerant, what could one expect from the non-tolerant genotype?

3. Was there other members of AP2 domain family also showed chnages in gene expression under water logging stress?

4. In the section 2.2 of results, it was not sure what plant tissue was used to estimate the gene expression under different hormonal and stress conditions. 5. The AdRAP2.3 expression was highest in fruit related parts not in roots? What is the possible reason for this?

Figures are presented with (A), (B) and (C) sub-headings, but figure legends lack the information. Even it seems like some of the figures legends are missing in my version of manuscript. They should be corrected. 

6. I can understand that this gene provides tolerance under water logging stress in transgenic tobacco plants? But again, what will be the use of this gene in kiwifruit because it has been isolated from kiwifruit? Do the expression of this gene can explain variation in water logging tolerance in diverse kiwifruit germplasm? I think that tracking the association of this gene with water logging tolerance in diverse kiwifruit genotypes will be most useful for kiwifruit production? This should be discussed in the manuscript. 

7. I see an error thoroughout the manuscript in place of references. It says "Error! Reference source not found.". Please make sure it will be corrected. 

Author Response

1. the kiwifruit is one of the most recently domesticated fruit crops. However, the majority of currently kiwifruit cultivars are susceptible to waterlogging stress. In many regions of China, especially in Eastern China, kiwifruit plants suffer from excess rainfall during the summer rainy season. Flooding affects fruit yield severely and trees even die, which restricts the development of the kiwifruit industry
2. Kiwifruit ‘Jinkui’ was tolerance to waterlogging,and understanding the mechanism of waterlogging stress responses is important to improve non-tolerant kiwifruit variety

3. The transcriptome sequencing analysis in ‘Jinkui’ under waterlogging stress showed that the family of AP2/ERF transcription factors contained 14 up-regulated and 14 down-regulated members in the treatment samples. And the transcription comp67160_0_seq1, which was ERF VIIs member named AdRAP2.3, was strongly upregulated in roots of Actinidia deliciosa during the first 96 h after waterlogging treatment
4. the expresion of AdRAP2.3 was detected in different tissues of kiwifruit wahic was not treatment by any hormonal and stress conditions.

5. Revised.
6. It is of great significance to introduce the gene into the kiwifruit varieties with poor waterlogging resistance in future and cultivate the kiwifruit excellent germplasm resources with strong waterlogging resistance to improve the quality and increase the efficiency of kiwifruit.

7. Revised.

Reviewer 2 Report

1. In this manuscript, authors reported a novel ethylene response factor from kiwifruit, AdRAP2.3, and discovered that this gene might be function in respond to waterlogging stress through overexpression analysis. The experimental design and results in this study could partly support the conclusion. However, authors should make more efforts on manuscript writing, especially on the ‘Discussion’ section. I cannot find any discussion content in ‘Discussion’ section. Authors only descripted backgrounds and repeated results again. The purpose of the discussion is to interpret and describe the significance of your findings in light of what was already known about the research problem being investigated, and to explain any new understanding or insights about the problem after you've taken the findings into consideration. It’s highly recommended authors could re-write discussion section.
2. In addition, it’s better to include a ‘Conclusion’ section after Discussion, which could help readers to find out the main findings of this study.
3. Except these writing technical, English writing in this manuscript also need to be improved and polished, as too much typo and grammar errors were found.  Some suggestions are listed below to help with manuscript improvements.

3.1.     When you talk about gene names, italic form like ‘AdRAP2.3’ could be used. However, when you talk about name of transcription factor (protein), ‘AdRAP2.3’ should be used. Please correct these errors in the text and figures.

3.2.     Line 36-39, please add references for these sentences.

3.3.     Line 95 and figure 2b, authors state expression level reduced ‘slightly’ at 48h. It’s not clear. If compared to 0h, it seems there was a little bit increase at 48h from figure. If compared to 4h, the decrease cannot be described as ‘slightly’.

3.4.     Line 153, Fig. 8B should be changed to Fig. 7B. Please also notice that ‘Figure’ and ‘Fig.’ should be unified through the manuscript, and ‘b’ and ‘B’ should be unified in the subtitle of each figure.

3.5.     Figure 2-4, please explain the means of 0, 4, 12, 48 h in figure legends.

3.6.     Figure 5-7, please explain the means of WT, #1, #14, #26 in figure legends. Figure 6, please also explain FW and DW in legend.

3.7.     Figure 8 and 9, please explain CK and WT in legends.

3.8.     Please explain why only transgenic line #14 was used for enzymes activities analysis and down-stream gene expression analysis.

3.9.     Line 187-188, ‘There are no significantly difference…… in roots’. Did this happen under control condition?

Author Response

1. The discussion part was re-write, see manuscript.

Waterlogging or submergence caused O2 deprivation in the soil [25, 26]. ERF members are important regulators of low oxygen tolerance extensively studied in many plants [8,10,11]. The ERF family plays a crucial role in the determination of survival of Arabidopsis and rice, which could reduce oxygen availability[11,27]. Previous studies showed that there are five ERF VIIs genes in Arabidopsis, two ERF VIIs genes (HYPOXIA RESPONSIVE ERF1/2) were greatly enhanced at the transcriptional and translational levels by O2 deprivation at multiple developmental stages, and the other three ERF VIIs genes were constitutively expressed (RAP2.12, RAP2.2, and RAP2.3) and further upregulated by darkness or ethylene in A. thaliana [11, 28, 29]. Transgenic Arabidopsis plants overexpressing HRE1 showed an improved tolerance of anoxia [11]. The transcriptome sequencing analysis showed that there are 28 AP2/ERF transcription factors regulated by waterlogging in kiwifruit. In this study, A. deliciosa AdRAP2.3 gene was induced significantly by waterlogging, and overexpression of the AdRAP2.3 gene conferred waterlogging stress tolerance in transgenic tobacco plants. Under completely submerged, survival rate, fresh weight and dry weight of transgenic tobacco lines were higher significantly than those of WT. These results suggested that kiwifruit AdRAP2.3 play key role in resistance to waterlogging.

25. Bailey-Serres, J.; Voesenek, L.A. Flooding stress: Acclimations and genetic diversity. Ann. Rev. Plant Boil. 2008, 59, 313–339.

26. Yamauchi, T.; Watanabe, K.; Fukazawa, A.; Mori, H.; Abe, F.; Kawaguchi, K.; Oyanagi, A.; Nakazono, M. Ethylene and reactive oxygen species are involved in root aerenchyma formation and adaptation of wheat seedlings to oxygen-deficient conditions. J. Exp. Bot. 2014, 65, 261–273.

27. Licausi, F.; Van Dongen, J.T.; Giuntoli, B.; Novi, G.; Santaniello, A.; Geigenberger, P.; Perata, P. HRE1 and HRE2, two hypoxia-inducible ethylene response factors, affect anaerobic responses in Arabidopsis thaliana. Plant J. 2010, 62, 302–315.

28. Gibbs, D.J.; Bacardit, J.; Bachmair, A.; Holdsworth, M.J. The eukaryotic N-end rule pathway: Conserved mechanisms and diverse functions. Trends Cell Biol. 2014, 24, 603.

29. Licausi, F.; Pucciariello, C.; Perata, P. New role for an old rule: N-end rule-mediated degradation of ethylene responsive factor proteins governs low oxygen response in plants. J. Integr. Plant Boil. 2013, 55, 31–39.

2. In summary, the results showed that the increase of AdRAP2.3 expression during waterlogging stress was much higher than that during other environmental stresses and that the kiwifruit AdRAP2.3 gene is required during waterlogging stress. Upon root submerged, transgenic tobacco lines grew aerial root earlier than WT. Overexpression of AdRAP2.3 in transgenic tobacco improved the activities of PDC and ADH enzyme, and the expression levels of waterlogging mark genes NtPDC and NtADH in roots under waterlogging treatment. Overall, these results suggested that AdRAP2.3 might play an important role in resistance to waerlogging through regulation of PDC and ADH genes in kiwifruit. It is of great significance to introduce the gene into the kiwifruit varieties with poor waterlogging resistance in future and cultivate the kiwifruit excellent germplasm resources with strong waterlogging resistance to improve the quality and increase the efficiency of kiwifruit.

Expression of the AdPDC1 gene was down-regulated by ABA in kiwifruit, and overexpression of the AdPDC1 gene in Arabidopsis inhibited seed germination and root length under ABA treatment, indicating that ABA might negatively regulate the AdPDC1 gene under waterlogging stress.

3.1.    revised.

3.2.    revised

3.3.    revised

3.4.    revised

3.5.    Thank you very much. Means of 0, 4, 12, 48 h were explained in Figures.

3.6.    revised

3.7.    revised

3.8.   three transgenic tobacco lines were all improved to waterlogging, and has same results, so we selected only transgenic line #14 for enzymes activities analysis and down-stream gene expression analysis.

3.9.   Thank you very much. There are no significantly difference between WT and transgenic lines on transcript levels of NtPDC, NtADH, NtHB1, NtHB2 and NtPCO1 in roots under control condition.

Round  2

Reviewer 1 Report

Most of the comments have been addressed. However, I still think that authors should explore further the role of this gene in wide kiwifruit germplasm, which will be useful for its improvement under waterlogging conditions.  

Author Response

In future studies we will concentrate on introducing the AdRAP2.3 gene into the kiwifruit varieties with poor waterlogging resistance, such as ‘Hongyang’ to determine if this can enhance resistance to waterlogging, and to cultivate the kiwifruit excellent germplasm resources with strong waterlogging resistance to improve the quality and increase the efficiency of kiwifruit.

Reviewer 2 Report

The manuscript was highly improved after revise, and authors answered most of question. However, there are few suggestion still need to be considered.

1. When you talk about gene names, italic form like ‘AdRAP2.3’ could be used. However, when you talk about name of transcription factor (protein), ‘AdRAP2.3’ should be used. Please correct these errors in the text and figures. Obviously authors didn't carefully check all this issue. Please check line 14 and Figure 8.

2. Please explain why only transgenic line #14 was used for enzymes activities analysis and down-stream gene expression analysis.

Response: three transgenic tobacco lines were all improved to waterlogging, and has same results, so we selected only transgenic line #14 for enzymes activities analysis and down-stream gene expression analysis.

I can't agree with this response. Authors proved all three transgenic lines were improved to resistance of waterlogging stress, the followed enzymes activities analysis and gene expression analysis could help to illustrate why this function is improved from biochemical and molecular levels. I could not understand why the other lines were excluded from two important analysis. Additionally author didn't explain why line #14 was selected.

Author Response

1. Thank you very much for your attention about our carelessness, we have checked again.

2. All three transgenic lines were improved to resistance of waterlogging stress, and have similar results. And the result of transgenic line #14 to improve waterlogging was medium for all three transgenic. So we selected only transgenic line #14 for enzymes activities analysis and down-stream gene expression analysis.